# What to Do if the qPCR Test for SARS-CoV-2 or Other Pathogen Lacks Endogenous Internal Control? A Simple Test on Housekeeping Genes

**DOI:** 10.3390/biomedicines11051337

**Published:** 2023-05-01

**Authors:** Aleksandra Kuzan, Ivo Tabakov, Lukasz Madej, Anna Mucha, Lukasz Fulawka

**Affiliations:** 1Molecular Pathology Centre Cellgen, 50-353 Wroclaw, Poland; 2Department of Biochemistry and Immunochemistry, Wroclaw Medical University, 50-367 Wroclaw, Poland; 3Faculty of Biotechnology, University of Wroclaw, Joliot-Curie 14a, 50-383 Wroclaw, Poland; 4Collegium Medicum, Jan Kochanowski University, 25-516 Kielce, Poland; 5Department of Genetics, Wroclaw University of Environmental and Life Sciences, 51-631 Wroclaw, Poland

**Keywords:** COVID-19, SARS-CoV-2, diagnostics, false-negative results, PCR

## Abstract

Some of the products for the molecular diagnosis of infections do not have an endogenous internal control, and this is necessary to ensure that the result is not a false negative. The aim of the project was to design a simple low-cost RT-qPCR test that can confirm the expression of basic metabolism proteins, thus confirming the quality of genetic material for molecular diagnostic tests. Two successful equivalent qPCR assays for the detection of the GADPH and ACTB genes were obtained. The course of standard curves is logarithmic, with a very high correlation coefficient R2 within the range of 0.9955–0.9956. The reaction yield was between 85.5 and 109.7%, and the detection limit (LOD) with 95% positive probability was estimated at 0.0057 ng/µL for GAPDH and 0.0036 ng/µL for ACTB. These tests are universal because they function on various types of samples (swabs, cytology, etc.) and can complement the diagnosis of SARS-CoV-2 and other pathogens, as well as potentially oncological diagnostics.

## 1. Introduction

Thanks to vaccination against the severe acute respiratory syndrome coronavirus 2 (SARS-CoV-2), the rate of development of the epidemic has clearly slowed down. However, summarizing the statistics covering 3 years of the pandemic, the World Health Organization (WHO) has reported over 753 million confirmed cases and over 6.8 million deaths globally [1]. It should be emphasized that these numbers include only confirmed cases because in the case of coronavirus disease 2019 (COVID-19), a large proportion of patients were asymptomatic and did not come to the examination or did not have access to testing, so the actual incidence is underestimated [2]. Moreover, due to the severity of the pandemic, test kits were developed and approved quickly to meet the worldwide demand for large-scale tests. Not all kits were equipped with an appropriate endogenous or exogenous internal control. In the case of this disease, it is particularly important to diagnose the infection quickly and effectively because the basic strategy to fight the spread of the virus, due to the lack of specific effective treatment, was to isolate patients.

Meanwhile, even 58 in 1000 tested patients may have received a false-negative result of the SARS-CoV-2 real-time reverse transcriptase polymerase chain reaction (RT-PCR) test during the pandemic [3]. Some studies report that as many as 25% of evaluated laboratories incorrectly interpreted the external quality assessment exercise samples, negatively matching the standards required [4]. This effect may be mainly due to one of the following reasons: the swab was taken before viral replication, the swab was taken incorrectly or the patient was not properly prepared for the procedure, or it was the result of contamination, the presence of amplification inhibitors, mutations in the viral genome or operator errors [3,5]. Due to the significant possibility of infecting others after receiving a false-negative result, we conclude that the risk of laboratory and pre-laboratory errors should be minimized. The strategy to evaluate the correctness of sampling is the application of an endogenous internal control. This procedure involves the parallel amplification and detection of viral and human housekeeping genes (HKGs). In some kits, such control is provided by the manufacturer. The remaining ones contain an exogenous internal control, i.e., synthetic nucleic acid, which is amplified in parallel with viral one. Thirty-four percent of PCR kits in Poland detecting SARS-CoV-2 have an endogenous control. In addition, 5% of tests have both an endogenous and exogenous internal control. The remaining 61% of these tests cannot demonstrate whether human cells were collected in sufficient amounts [5]. The aim of this project was to design a simple and cost-effective test for verification of the quality of genetic material, independently of the commercial test.

An ideal gene, used as a reference gene in the RT-PCR method, should be characterized by stable expression in all cells of the body, independent of age and metabolic status, including independence from cancer and the influence of hormones and xenobiotics [6]. The genes most often selected for the design of internal controls are 18S rRNA (18S ribosomal RNA), 28S rRNA (28S ribosomal RNA), TUBA (α-tubulin), ACTB (β-actin), β2M (β2-microglobulin), ALB (albumin), RPL32 (ribosomal protein L32), TBP (TATA sequence binding protein), CYCC (cyclophilin C), EF1A (elongation factor 1α), GAPDH (glyceraldehyde-3-phosphate dehydrogenase), HPRT (hypoxanthine phosphoribosyl transferase), RPII (RNA polymerase II) and RNAse P [7,8]. ACTB and GAPDH were selected for this project. The choice was made due to the fact that these genes belong to the classic genes of basic metabolism, historically considered to be expressed stably in human cells [9], and are already used as reference genes in molecular testing of nasopharyngeal and oropharyngeal swabs by some scientists [10].

## 2. Materials and Methods

The starting material was Universal Human Reference RNA (ThermoFisher Scientific, Waltham, MA, USA) and RNA isolated from swabs collected for SARS-CoV-2 diagnostics (positive/negative/non-diagnostic) and other biological materials (cheek swab, saliva and cervical swabs).

The paper describes the procedure and results of this work, the primary aim of which was diagnostic use. According to Polish law, standardization and laboratory validation do not require evaluation by the institutional research ethics committee. No human material was used beyond diagnostic purposes. No patient data were used in this work.

### 2.1. RNA Isolation in the Study Was Conducted on Remnants of Nucleic Acid Isolates 

RNA isolation was performed using commercial CE-IVD kits, according to the manufacturers’ instructions. Nucleid acids from nasopharyngeal and pharyngeal swabs were isolated with the EliGene^®^ Viral RNA/DNA FAST Isolation Kit (Elisabeth Pharmacon Ltd., Brno-Zidenice, Czech Republic), TANBead^®^ Nucleic Acid Extraction Kit (Taiwan Advanced Nanotech Inc., Taoyuan City, Taiwan) or ANDiS Viral RNA Auto Extraction & Purification Kit (3D Biomedicine Science & Technology Co., Ltd., Shanghai, China). Cheek swabs, saliva and cervical swabs were isolated using Invisorb Spin Virus DNA (AmpliSens, Moscow, Russia) or the Invisorb^®^ Spin Universal Kit (Invitek Molecular GmbH., Berlin, Germany).

### 2.2. Reverse Transcription 

To obtain cDNA from RNA, reverse transcription was performed using the First Strand cDNA Synthesis Kit for RT-qPCR (ThermoFisher Scientific, Waltham, MA, USA), preceded by removing contaminating genomic DNA from isolate by using dsDNase included with the kit. The reaction mix was prepared according to the manufacturers’ instructions: 1 µL dsDNase buffer, 1 µL dsDNase, RNase-free water and isolate sample depending on the amount of template RNA. After incubation with dsDNase, the following reagents were added: 4 µL 5× reaction mix, 2 µL Maxima Enzyme Mix and 4 µL RNase-free water. The reverse transcription reaction was carried out by incubation for 10 min at 25 °C followed by 15 min at 50 °C.

### 2.3. Real-Time PCR 

The qPCR reaction was performed with the QuantiFast SYBR Green PCR Kit (Qiagen, Düsseldorf, Germany) and Bio-Rad CFX96 instrument (Bio-Rad Laboratories, Inc., Hercules, CA, USA). 

Based on our experience in work with RNA, we selected two housekeeping genes: beta actin (ACTB) and glyceraldehyde-3-phosphate dehydrogenase (GAPDH).

Primers: ACTB forward: GGCCAACCGCGAGAAGA;ACTB_reverse: CCGTGGTGGTGAAGCTGT;Product length: 272 bp;GAPDH_forward: CATGAGAAGTATGACAACAGCCT;GAPDH_reverse: AGTCCTTCCACGATACCAAAGT;Product length: 205 bp and 113 bp.

Subsequently, reactions of mixtures consisting of 5 µL QuantiFast SYBR Green PCR Master Mix, 0.5 µL F primer, 0.5 µL R primer, 3 µL RNase-free water and 1 µL template DNA were performed. The reaction was carried out according to the conditions presented in Table 1.

### 2.4. Preparation of Standard Curve

The solutions of cDNA were produced based on the following concentrations of reference RNA to prepare the standard curve: 8 ng/µL; 2 ng/µL; 0.5 ng/µL; 0.125 ng/µL; and 0.03125 ng/µL.

To estimate the repeatability of the method, the standard deviation (*SD*) and the coefficient of variation (*CV*) between replicates were calculated for several replicate samples, according to the formulas below.
$$SD=\sqrt{\frac{\sum_{i-1}^{n}{\left(xi-\bar{x}\right)}^{2}}{n-1}}$$
$$CV=\frac{SD}{\bar{x}}*100\%$$
where *SD*—standard deviation; *CV*—coefficient of variation; µ—the mean of all results in the group; and σ-standard deviation for all results in the group.

### 2.5. Electrophoresis

The gel was prepared by dissolving 2 g of agarose in 100 mL TBE buffer followed by the addition of either 5 µL Syngen GreenDNA Gel Stain (Syngen, Wrocław, Poland) or 5 µL EurX Simply Safe™ (EurX, Gdańsk, Poland). A total of 5 µL of each sample was mixed with 1 µL Qiagen GelPilot loading buffer (Qiagen, Düsseldorf, Germany) and loaded into the wells. For reference, 6 µL of EurX perfect DNA 100 bp–1000 bp ladder was used on both outer wells. Electrophoresis was run for 40 min at 120 V.

Various electrophoresis experiments had shown similar results in the frame of expected values:-cDNA GAPDH around 205 bp and 113 bp.-cDNA ACTB around 272 bp.

### 2.6. Comparison with the Gold Standard

The developed assay was compared with the gold-standard assay—TaqMan ACTB Gene Expression Assay (assay ID: Hs01060665_g1) (Thermo Fisher Scientific, Waltham, MA, USA)—performed with TaqMan Fast Advanced Master Mix (ThermoFisher Scientific, Waltham, MA, USA). The comparison was conducted on 24 cDNA samples obtained from remnants of nucleic acid isolates from current diagnostics, as well as remnant isolates stored at −20 °C in the Molecular Pathology Centre Cellgen Biobank (the oldest materials have been stored for up to 5 years in controlled conditions). 

### 2.7. Limit of Detection (LoD) Determination

For evaluation of the LoD, probit analysis was conducted using data from previously performed qPCR on a series of dilutions for both GAPDH and ACTB genes.

## 3. Results

Examples of plots of fluorescence increase over time are shown in Figure 1A. As predicted, the logarithmic course of the curves is observed. According to the guidelines of the EMEA (European Medicines Agency) and FDA (Food and Drug Administration), the CV for diagnostic methods should be less than 15% [11,12]. In the case of our tests, the CV is in the range of 0.14–1.66%, which is very low, i.e., the desired result. The conclusion is that the repeatability of the method is very high.

The specificity of the products is estimated after analysis of the melting curve (Figure 2A,B). The melting temperatures were 87.5 °C for ACTB and 82.0 °C for the GAPDH gene. We conclude that the products of both PCR reactions are specific.

Standard curves were determined from the obtained data from the experiment, taking into account increasing cDNA concentrations. An example of the curves for both genes is provided in Figure 3A,B. Efficiency is observed in the range of 85.5–109.7%, which is a satisfactory result.

The coefficient of determination, R2, the linearity parameter, is satisfied and is 0.996–0.997 for actin (cDNA) and 0.995–0.998 for GAPDH (cDNA). 

The sensitivity for GAPDH was 80%, and for ACTB, 63%. When analyzing both genes collectively (a positive result was determined as the presence of amplification in at least one gene), sensitivity increased to 92%. 

The LoD with 95% positive probability was estimated at 0.0057 ng/µL for GAPDH with a confidence interval of 0.00257, 0.0206 at 95% probability and 0.0036 ng/µL for ACTB with a confidence interval of 0.00176, 0.00853 at 95% probability (confidence level data collected in Table 2). Data used for the calculation of the probit curve are shown in Table 3 and Table 4 for GAPDH and ACTB, respectively. It is important to note that the actual concentration of RNA (cDNA) in the samples subjected to qPCR is 20-fold lower than the one given. It is due to diluting the sample during RT-PCR (reverse transcriptase PCR). These data are graphically presented in Figure 4A,B.

## 4. Discussion

The use of the real-time RT-PCR method, which allows monitoring of the process of amplification of genetic material in real time, enables quick and accurate measurement of the amount of specific genes present in the sample. During the subsequent stages of sample preparation aiming to determine the number of transcripts of the tested genes, changes in the amount of genetic material may occur. The most common cause is the different efficiencies of mRNA isolation and reverse transcription in individual samples. In such cases, it is difficult to discuss abnormalities such as overexpression or underexpression, which are the bases for diagnostics, for example, for cancer, without comparing the tested gene to the housekeeping gene. The HKG expression should be, by definition, constant [13]. For some analyses of swabs and other samples from patients for diagnostic purposes, the result may not necessarily be quantitative but qualitative enough, such as in the diagnosis of SARS-CoV-2 and other infections. Here, there is a relevant question of whether there is genetic material in the sample in quantity and quality that allows reliable molecular testing.

It is well known that although RT-PCR has high analytical specificity resulting in a minimal false-positive rate, its diagnostic sensitivity remains suboptimal, leading to the possibility of false-negative results. The time window in which false negatives are more common is not clear [3]. It seems to depend on the individual differences of patients, especially in the context of the state of their immune system.

The experiences of the diagnostic institution are described in our previous study—as many as 0.04% of results were non-diagnostic, using the tests containing an exogenous internal control [2]. On a scale of several hundred thousand tests performed, this is a significant number. The factors underlying the non-diagnostic results obtained with an exogenous internal control are inhibitors of PCR or, sometimes, errors in preparing the reaction. However, these tests give potentially false negatives in cases with inadequate matrix quality in the sample.

During the COVID-19 pandemic, especially in the first few months of its duration, there was a problem with the availability of kits for RNA isolation and qPCR for SARS-CoV-2 detection. Most of the tests available in Poland at that time contained an exogenous internal control.

This project achieved the intended goal: a simple economic test for the analysis of housekeeping genes in a variety of biological materials was carried out. Two genes were selected. The reason for this selection is described below.

The GAPDH gene produces a constitutive protein involved in many basic cellular functions such as glycolysis, DNA replication, nuclear RNA export, repair and apoptosis [14]. GAPDH is one of the best reference genes, but it is not ideal, as the expression of this gene is not identical in every tissue. For the skeletal muscle tissue and breast cells, a 14-fold difference in expression was observed. Moreover, the expression level of GAPDH can vary in diseases [7]. For throat swabs, however, it does not seem to matter, especially since we do not expect a quantitative result but a qualitative one, which indicates the presence of human RNA in the analyzed material. Other authors, for example, Rojas-Serrano or Wong et al., also believe that the gene serves as a good internal endogenous control to identify the possibility of obtaining false-negative results in the diagnosis of SARS-CoV-2 [14]. 

The ACTB gene encodes the protein ß-actin, a highly conserved cytoskeleton protein [15]. It is a protein essential in maintaining the shape of the cell and the segregation of chromosomes [6]. Although it has functioned as a reference gene for many years, there are reports that gene expression is unequal in various tissues [15]. However, it is one of the few proteins whose expression is considered to be independent of the patient’s age [6]. In the context of COVID-19 diagnostics, it is not a popular gene. The keywords “ACTB internal control SARS-CoV-2” entered in PubMed give no results. Nevertheless, some commercial companies use ACTB detection in diagnostic tests targeting SARS-CoV-2, e.g., the SARS-CoV-2 Virella test (Gerbion GmbH & Co. KG., Kornwestheim, Germany). Our data show that the ACTB-targeted test is sufficient to support COVID-19 diagnosis.

In this project, we compared two HKGs. The results indicate that both genes are equivalent, and we cannot distinguish the better one for internal control purposes. It is worth noting that although many studies have reported the use of only one HKG, a normalization strategy based on a single reference gene can lead to erroneous results. Therefore, some suggest using at least two reference genes to avoid significant errors [9]. 

This protocol uses a popular dye for the real-time reaction, SYBR Green I, which binds to double-stranded DNA, making its fluorescence increase almost 1000-fold [6]. The downside of this system is the non-specificity of binding. It results in the inability to analyze several genes in one reaction mixture. Moreover, non-specific sequences can form double-stranded structures (e.g., primer dimers or non-specific product amplification) and thus can lead to an increase in the fluorescence signal.

To avoid it, in addition to the fluorescence growth curve, a melting curve is analyzed at the end of the reaction with SYBR Green I, which allows us to determine the presence of amplicons of a specific length and nucleotide composition. Another solution is a more specific analysis using molecular probes that anneal to a particular sequence on the template. However, this approach significantly increases the cost of the test. In this respect, we have chosen SYBR Green I with melting curve verification. We believe it is optimal for simple diagnostic methods and sufficient enough to consider the method reliable.

The SARS-CoV-2 pandemic, among the numerous losses it caused, also brought a positive phenomenon: the dynamic development of biotechnology, especially in the area of molecular biology. Currently, various complex diagnostic tests are being designed based on loop-mediated isothermal amplification (LAMP), lateral flow immunoassays (LFIA), chemiluminescence-based immunoassays (CLIA), Clustered Regularly Interspaced Short Palindromic Repeats (CRISPR)-mediated technology and others. Among these various tests, it is worth noting, for example, the test developed by Maglaras et al. Their product is an LFA (lateral flow assay), where detection is visual. It does not require RNA isolation but is based on RT-PCR, and it requires an RT-PCR amplification step prior to the strip analysis [16]. Of the RT-LAMP tests, it is worth paying attention to the one prepared by Aldossary et al. This test is quick and requires no specialized equipment (only a heat block or water bath to maintain a temperature of 65 °C), and after simple modification, saliva can be used instead of swab material [17]. The disadvantage is that there is no internal control for each test, which affects its reliability. Of the many CRISPR-based assays targeting COVID-19, one can highlight assays developed, for example, by Xinjie Wang et al. [18] and Rui Wang et al. [19]. However, the authors of the review work on CRISPR-based COVID-19 testing conclude that it is not yet possible to replace RT-qPCR because each test they describe has potentially reduced sensitivity compared to the gold standard. Nevertheless, this does not undermine its many advantages, the most significant of which is its usefulness for point-of-care testing (POCT) [20]. 

We also recommend the review paper by Li et al., in which the authors summarize modern-smartphone-based detections for SARS-CoV-2 [21]. However, the authors themselves point out that the price and complexity of the smartphone-based detection methods still need to be further optimized [21]. 

We would like to emphasize that the protocol we developed belongs to the classic, well-known methodologies of molecular biology, such as the polymerization chain reaction, enjoying the great trust of scientists and laboratory diagnosticians. It does not have the advantages of, for example, the cassette test, but it is characterized by basically absolute reliability. This fact, along with the relative simplicity of the methodology and cost, are the main advantages of such a protocol. We mark that it is not a stand-alone test for diagnosing COVID-19, but it was created from the need to supplement commercial tests targeting SARS-CoV-2 genes to compensate for the lack of an internal endogenous control in some products and to make the results more credible.

A separate issue is the need to verify the correctness of the RT-PCR reaction by performing an exogenous internal control. This topic is not the subject of our work, but we pay attention to analyzing the need for an additional step to authenticate the method. An example of a publication describing the protocol for the preparation of such a control is the work of Kavlick [22].

To summarize, the test prepared by us is an endogenous internal control that can be conducted independently of the commercial SARS-CoV-2 test and that is relatively cheap, simple and characterized by good validation parameters. The implementation of this test in laboratory practice can improve the quality of laboratory diagnostics and contribute to faster control of the COVID-19 epidemic. It is also universal enough to be useful for any type of molecular test targeting a variety of viral and bacterial infections.

## Figures and Tables

**Figure 1 biomedicines-11-01337-f001:**
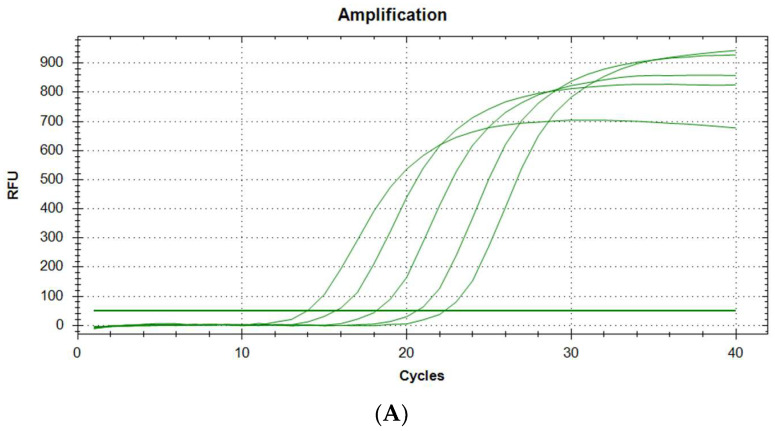
(**A**). Amplification plots of GAPDH cDNA in qPCR (5 samples with increasing cDNA concentration). (**B**). Amplification plots of ACTB cDNA in qPCR (5 samples with increasing cDNA concentration).

**Figure 2 biomedicines-11-01337-f002:**
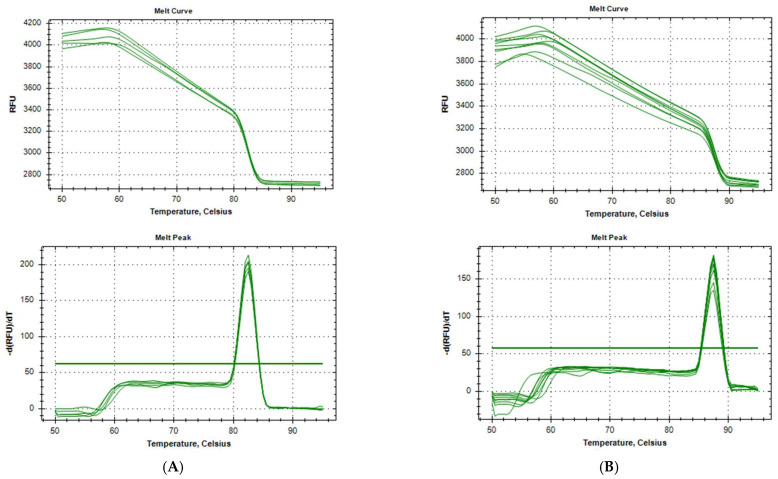
(**A**). Melt curve (**A**) and its first minus derivative (**B**) from qPCR of the GAPDH gene. (**B**). Melt curve (**A**) and its first minus derivative (**B**) from qPCR of the ACTB gene.

**Figure 3 biomedicines-11-01337-f003:**
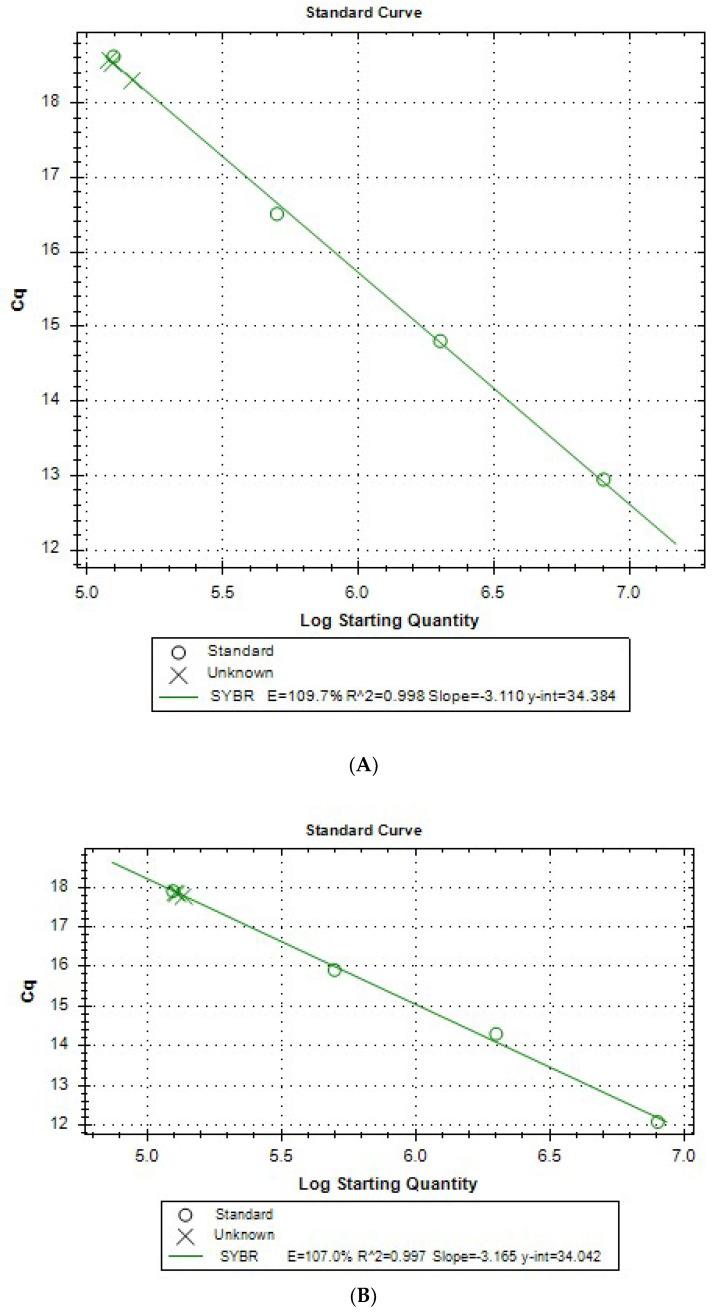
(**A**). Four-point calibration curve for GAPDH. (**B**). Four-point calibration curve for ACTB.

**Figure 4 biomedicines-11-01337-f004:**
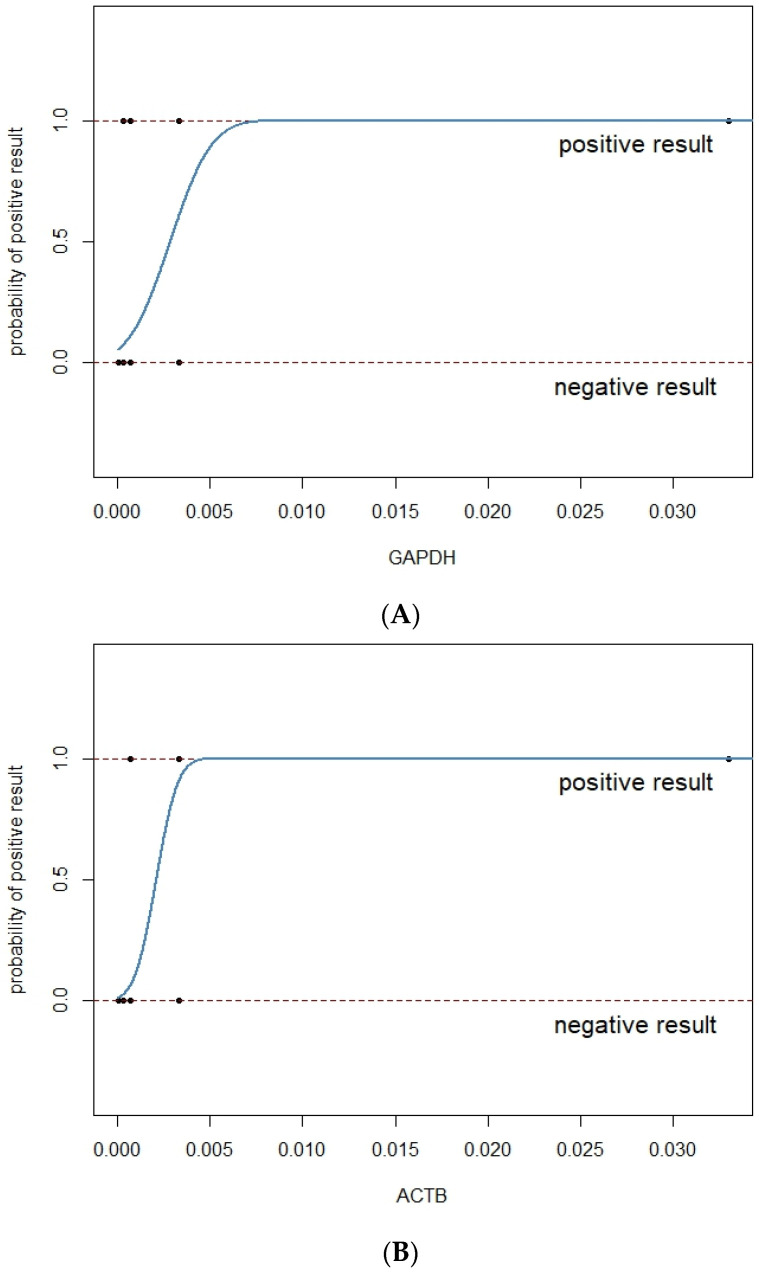
(**A**). Probit curve used for estimation of LoD for GAPDH. (**B**). Probit curve used for estimation of LoD for ACTB.

**Table 1 biomedicines-11-01337-t001:** PCR reaction conditions.

Stage	Temperature (°C)	Time	
initial denaturation	95	5′	
denaturation	95	10″	amplification 40 cycles
annealing and elongation	60	30″	
performing a melting curve	50–95 °C, increase by 0.5 °C	5″	

**Table 2 biomedicines-11-01337-t002:** Data for confidence levels in reactions for GAPDH and ACTB.

	2.5%	97.5%
(Intercept)	−2.586384	−0.8339351
GAPDH	205.169616	966.1144778
(Intercept)	−3.693924	−1.301388
ACTB	625.699910	1678.059262

**Table 3 biomedicines-11-01337-t003:** Data used for calculation of probit curve for GAPDH.

Concentration of RNA in Sample (ng/µL)	Number of Replicates	Number of Positive Tests
0.033	3	3
0.0033	10	6
0.00067	10	1
0.00033	10	1
0.000033	3	0

**Table 4 biomedicines-11-01337-t004:** Data used for calculation of probit curve for ACTB.

Concentration of RNA in Sample (ng/µL)	Number of Replicates	Number of Positive Tests
0.033	3	3
0.0033	10	9
0.00067	10	1
0.00033	10	0
0.000033	3	0

## Data Availability

Data supporting the reported results can be obtained from the corresponding author upon request.

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
