# Peer review of "What to Do if the qPCR Test for SARS-CoV-2 or Other Pathogen Lacks Endogenous Internal Control? A Simple Test on Housekeeping Genes"

_biomedicines, 2023, doi:10.3390/biomedicines11051337_

Round 1

Reviewer 1 Report

Congratulations for your superb work, the novelty of your finding may assist other colleagues to choose new sufficient methods to approach pathogens.

My only concern is regarding the reproducibility of the method examined and the strengths or weakness in comparison to analogous published PCR modifications.

Author Response

We thank the Reviewer for the positive initial evaluation of our manuscript. We hope that after completing and correcting the manuscript in response to the comments of the Reviewers and the Editor, the manuscript will turn out to be even better. Below is a direct response to the Reviewer's comments.

Congratulations for your superb work, the novelty of your finding may assist other colleagues to choose new sufficient methods to approach pathogens.

My only concern is regarding the reproducibility of the method examined and the strengths or weakness in comparison to analogous published PCR modifications.

The use of our methodology relative to commercial and other kits for amplification of ACTB and GAPDH genes, is much cheaper than these kits. No commercial kit for assessing the quality of material with hausekeeping gene expression examination is based on two-gene testing.  In commercial kits for measuring gene expression, the sequence of primers and probes is not known and in effect the amplified region.

Our paper is „utilitarian” since it presents results of the internal development of assay for practical purposes, to be used only in our lab as an ancillary test. It is not required by any recommendation, when diagnostic procedures are conducted strictly according to CE-IVD kits manuals provided by producers. It has to be underlined that our laboratory has always achieved the highest marks in external quality control programs, even before the development of this procedure. The aim of development of the ancillary procedure was rather to exceed quality of our diagnostic services.

We strongly believe that publication of the developed procedure, which is cheap and easy to introduce into routine work, would help wider community to increase the quality of diagnostic services and to lower the number of false negative results.

Thanks again to the Reviewer for suggestions, we hope that we have responded satisfactorily and that the manuscript will be accepted for publication in Biomedicines in Special Issue „State-of-the-Art Molecular and Translational Medicine in Poland”.

Reviewer 2 Report

The manuscript by Kuzan et al tried to provide an alternative method to detect house-keeping genes for SARS-CoV-2 detecting qPCR test lacking endogenous control. My major concerns are as follows:

1. Dozens of RT-qPCR assays for detecting house-keeping genes have been developed, what’s the difference between the developed assays and the previous assays? Could we use previously developed assays to support the qPCR assays lacking endogenous control?

2. How about the clinical performances of the developed assays comparing to a gold-standard assay?

3. Rather than described the LoD of the developed assays as a wide range, statistical method such as Probit regression analysis should be applied to determine the exact LoD.

Author Response

We would like to thank the Reviewer for the time devoted to the analysis of this work, valuable comments and the opportunity to improve the work. Below the answers "point by point".

The manuscript by Kuzan et al tried to provide an alternative method to detect house-keeping genes for SARS-CoV-2 detecting qPCR test lacking endogenous control. My major concerns are as follows:

  1. Dozens of RT-qPCR assays for detecting house-keeping genes have been developed, what’s the difference between the developed assays and the previous assays? Could we use previously developed assays to support the qPCR assays lacking endogenous control?

  • Yes, we can use previously developed assays to support the qPCR assays lacking endogenous control. However, the assay developed by us has been internally validated on diagnostic samples and nucleic acid isolates obtained with methods used routinely in our lab. Another important reason for developing the assay is that the price is significantly lower than commercial assays. It enables us to check the quality of significantly higher number of samples tested, thus lowering the number of false negative results.

  1. How about the clinical performances of the developed assays comparing to a gold-standard assay?

  • The use of our methodology relative to commercial and other kits for amplification of ACTB and GAPDH genes, is much cheaper than these kits. No commercial kit for assessing the quality of material with hausekeeping gene expression examination is based on two-gene testing. In commercial kits for measuring gene expression, the sequence of primers and probes is not known and in effect the amplified region.

 Our paper is „utilitarian” since it presents results of the internal development of assay for practical purposes, to be used only in our lab as an ancillary test. It is not required by any recommendation, when diagnostic procedures are conducted strictly according to CE-IVD kits manuals provided by producers. It has to be underlined that our laboratory has always achieved the highest marks in external quality control programs, even before the development of this procedure. The aim of development of the ancillary procedure was rather to exceed quality of our diagnostic services. It would be economically unjustified to introduce full clinical validation, which seems to be suggested by reviewers.

The work has been prepared as a part of the local support programme aiming to increase competitiveness of small companies from the City of Wrocław. The programme does not support basic research.

We strongly believe that publication of the developed procedure, which is cheap and easy to introduce into routine work, would help wider community to increase the quality of diagnostic services and to lower the number of false negative results.

  1. Rather than described the LoD of the developed assays as a wide range, statistical method such as Probit regression analysis should be applied to determine the exact LoD.
  • We decided to replace a wide range into exact value of the LoD (0.003ng / µl for both genes) in the manuscript.

Thanks again to the Reviewer for all suggestions, we hope that we have responded satisfactorily and that the manuscript will be accepted for publication in Biomedicines in Special Issue „State-of-the-Art Molecular and Translational Medicine in Poland”.

Round 2

Reviewer 2 Report

I do not think the authors' reply have fitted my concerns. Although the cost of an assay is important, the quality or performance of the assay is also essential. Thus, the authors should carried out prerformance comparisoins between the developed assay and a gold-standard assay. Meanwhile, I failed to detect any description on the methods for LoD determination.

Author Response

We thank the Reviewer very much for suggestions for improving the manuscript. We apologize for taking so long to prepare a response, but we had to purchase reagents and perform a number of analyses to implement the Reviewer's suggestions. We performed both LoD determination using the Probit regression method and a comparison with the gold standard (Taq Mann Assay). We have included the results in the manuscript. We hope that the current version of the paper will be favorably reviewed by the Reviewer. Thank you again for your time and help in improving the paper.

Round 3

Reviewer 2 Report

I don not think that my privious concerns have been met.

1.  Although the developed assay has been applied in the author's lab, a comprehensive evaluation of the assay performance is requried before publication of the manuscript. Meanwhile, I could not understand the author's statement that their assay is much cheaper. Because application of other published assays rather than commercial kits is also cost effective.

2.  I do not agree with the author's opinion that this manuscript is only a utilization of a estabished assay. Rather, the assay should be developed by comparing with a golden standard assay. This is not a recommendation but a request.

3. About the LoD calculation of DNA quantitation, it is conventionally expressed by copies/ul, rather than ng/ul. Meanwhile, the it should indicate at which level of probability and show the confidence interval of the Probit regression analysis.

Author Response

We thank the Reviewer for his patience in developing our research paper. We have tried to make corrections and respond point by point to each comment.

  1. Although the developed assay has been applied in the author's lab, a comprehensive evaluation of the assay performance is requried before publication of the manuscript. Meanwhile, I could not understand the author's statement that their assay is much cheaper. Because application of other published assays rather than commercial kits is also cost effective.

We agree with the Reviewer that our assay is cost-comparable to other published assays. However, we believe that our protocol enriches the pool of practical knowledge available to other investigators. especially in Poland, where exactly the products we use in our laboratory are available (which seems important since we are applying for the special issue „State-of-the-Art Molecular and Translational Medicine in Poland”).

  1. I do not agree with the author's opinion that this manuscript is only a utilization of a estabished assay. Rather, the assay should be developed by comparing with a golden standard assay. This is not a recommendation but a request.

The developed assay was compared with the gold-standard assay. We refer to the part of the work entitled „2.6. Comparison with the gold-standard”.

  1. About the LoD calculation of DNA quantitation, it is conventionally expressed by copies/ul, rather than ng/ul. Meanwhile, the it should indicate at which level of probability and show the confidence interval of the Probit regression analysis.

We are using reference RNA that is not specified except for the given concentration, therefore we are unable to estimate number of copies in the sample. We can try to calculate it but the estimated value will be less credible than ng/uL due to the fact that we are not using synthesized oligo’s but an unspecified RNA. We believe that the concentration should be expressed by ng/uL in this particular case. Moreover, In our experience, many kits, even commercial ones, use pg (ng)/ul results, e.g. Investigator Quantiplex Kit (QIAGEN).

Summarizing, thanks to the Reviewer's comments, we had the opportunity to learn more deeply the basics of statistics and test comparison. Therefore, we would like to thank again for the Reviewer's contribution to improving the manuscript, and we look forward to your feedback.

Yours faithfully,

Authors